# A Critical Review on the Sensing, Control, and Manipulation of Single Molecules on Optofluidic Devices

**DOI:** 10.3390/mi13060968

**Published:** 2022-06-18

**Authors:** Mahmudur Rahman, Kazi Rafiqul Islam, Md. Rashedul Islam, Md. Jahirul Islam, Md. Rejvi Kaysir, Masuma Akter, Md. Arifur Rahman, S. M. Mahfuz Alam

**Affiliations:** 1Department of Electrical and Electronic Engineering, Dhaka University of Engineering & Technology, Gazipur 1707, Bangladesh; m.rahman@duet.ac.bd (M.R.); rafiqul@duet.ac.bd (K.R.I.); mrislam@duet.ac.bd (M.R.I.); masuma@duet.ac.bd (M.A.); marahman@duet.ac.bd (M.A.R.); 2Department of Electrical and Electronic Engineering, Khulna University of Engineering & Technology, Khulna 9203, Bangladesh; jahirul@eee.kuet.ac.bd; 3Department of Electrical and Computer Engineering, University of Waterloo, 200 University Ave. W, Waterloo, ON N2L 3G1, Canada; mrkaysir@uwaterloo.ca; 4Waterloo Institute for Nanotechnology, University of Waterloo, 200 University Ave. W, Waterloo, ON N2L 3G1, Canada

**Keywords:** single-molecule method, optofluidics, microfluidics, fluorescence, lab-on-a-chip, nanopore

## Abstract

Single-molecule techniques have shifted the paradigm of biological measurements from ensemble measurements to probing individual molecules and propelled a rapid revolution in related fields. Compared to ensemble measurements of biomolecules, single-molecule techniques provide a breadth of information with a high spatial and temporal resolution at the molecular level. Usually, optical and electrical methods are two commonly employed methods for probing single molecules, and some platforms even offer the integration of these two methods such as optofluidics. The recent spark in technological advancement and the tremendous leap in fabrication techniques, microfluidics, and integrated optofluidics are paving the way toward low cost, chip-scale, portable, and point-of-care diagnostic and single-molecule analysis tools. This review provides the fundamentals and overview of commonly employed single-molecule methods including optical methods, electrical methods, force-based methods, combinatorial integrated methods, etc. In most single-molecule experiments, the ability to manipulate and exercise precise control over individual molecules plays a vital role, which sometimes defines the capabilities and limits of the operation. This review discusses different manipulation techniques including sorting and trapping individual particles. An insight into the control of single molecules is provided that mainly discusses the recent development of electrical control over single molecules. Overall, this review is designed to provide the fundamentals and recent advancements in different single-molecule techniques and their applications, with a special focus on the detection, manipulation, and control of single molecules on chip-scale devices.

## 1. Introduction

Single-molecule (SM) techniques are relatively new endeavors with methods of probing and examining single biomolecules, usually one at a time, that reveal the details of individual biomolecules with heterogeneity and highly complex yet essential biological processes [1,2,3]. Conventional ensemble measurements involve the study of enormous biomolecules simultaneously and depict the statistical average of the biological processes that have a significant role in achieving understanding of the biological system with remarkable impacts in related fields [4,5,6,7,8]. However, the bulk measurements are mostly insufficient and sometimes unable to comprehend the processes happening at the molecular level, which probably plays the most crucial role in governing the biological systems [6,9]. The importance of SM techniques can probably be better expressed with the famous quote by Selvin and Ha “*Many biological reactions are too complex to fully comprehend through the use of conventional ensemble techniques. At the most fundamental level, all biological reactions occur* via *the action of single enzymes, DNA molecules, or RNA molecules. Studying one biological macromolecule at a time, or biology in singulo, can provide us with extraordinarily clear and often surprising views of these molecules in action*” [10]. SM studies provide information on short-lived states, heterogeneous and inhomogeneous properties of biomolecules, functional differences of individual molecules, dynamics in the molecular process, and so on [6,11,12,13]. Many fundamental questions associated with molecular biology, life science, and related fields are already well addressed using SM methods [14,15,16,17,18,19]. In many cases, SM approaches have reshaped the conventional way of thinking, led to new views and methodologies, and played a pioneering role in some ongoing influential topics such as personal diagnostics, precision medicine, medical product design, etc. [1,20,21,22,23]. Therefore, the growing expectation of better healthcare, diagnostics, a longer life span, and a better understanding of fundamental science leads to the conclusion to dig deeper and dive into the molecular levels with SM techniques. One way to fathom the tremendous appeal and growth of SM methods is to look at the exponential rate of research publications in related fields with the prediction that, shortly, most biological science publications will incorporate SM techniques in their studies [24]. Single-molecule experiments are developed to delve deeper into the molecular level and address individual molecules one at a time to characterize and understand the function of the building blocks of living organisms, therefore revealing the chemistry of life [2,25,26,27]. SM experiments have fused biology and life sciences with other branches of science such as physics, chemistry, engineering, etc. As the disciplines are quite different than each other (for example, the details of biological processes are not well known to physicists and engineers and vice versa), it requires knowledge and understanding of all the disciplines to devise the SM methods [27]. Nonetheless, the gap among these fields is getting narrower nowadays and more scientists and biologists are attracted to SM methods for obvious reasons. Therefore, SM methods are developing at a very fast rate, and new methods are proposed from diverse disciplines, making it difficult and almost impossible to cover all the aspects in a single review. Nonetheless, it is probably fair to say that optical methods, force-based methods, and electrical methods are the dominant SM experimental methods [1,14,28,29,30,31,32,33,34]. It is also quite common to exploit the advantages of the individual techniques for more functional integrated devices with added capabilities [14,35,36]. Due to obvious advantages, a number of SM techniques and devices have been developed by researchers around the world, and recently, the small, complete, chip-scale devices have gained much attention due to their enhanced capabilities, portable facilities, etc. [35,36,37,38,39,40,41]. Apart from SM imaging and detection, control and manipulation applications probably have received significant attention and huge efforts have been initiated in the scientific and lab-on-chip community for gaining insight into fundamental biological processes. These efforts are in turn leading to the creation of effective, affordable, miniaturized devices to meet the ever-increasing demand for improved understanding and biomedical devices. SM research is undoubtedly highly interdisciplinary, incorporating engineering disciplines, physics, chemistry, biology, material science, life science, medicine, proteomics, genomics, etc. Therefore, a full systematic review on SM research is difficult due to the vastness of the field. This review is designed to first provide insight into the fundamentals of microfluidic and optofluidic platforms with their integration capabilities and proceed toward the biosensing applications. Next, some popular single-molecule experimental methods are reviewed to gain better insight into the fundamentals and state of the art of single-molecule analysis with their capabilities and applications in diverse fields. In the subsequent sections, some applications of these methods are reviewed such as the sorting, manipulation, and trapping of single molecules and some reconfigurable optofluidic devices, with an emphasis on waveguide-based platforms. Finally, the electrical method, namely, nanopores and integrated nanopore devices, are reviewed with regard to the electrical sensing, integrated manipulation, and precise controlling of SM molecules, with an assessment of its outlook and probable future development scopes.

## 2. Microfluidics and Optofluidics

In the groundbreaking talk “*There is Plenty of Room at the Bottom*”, Richard Feynman shared his views on exploring and manipulating things at an ultra-small scale [42]. The speech has a tremendous influence on the transformation of technology and research and is sometimes considered as the inception of nanotechnology and miniaturization [43,44,45]. Miniaturization is a common trend in analytical applications that facilitates the creation of small, multifunctional, portable, low-cost, easy-to-use, reconfigurable devices, thereby gradually making them available to a higher percentage of the population [46,47,48,49]. Although the art of miniaturization is pioneered by the microelectronic industry, it has significantly reshaped other fields, opening numerous new possibilities. The development of microfluidics has followed the trend of microelectronics, exploiting the same fabrication methods to create miniaturized devices with micro-scale fluid-handling capabilities. Generally, microfluidic devices have at least one characteristic dimension of 100 µm or less [50]. For better visualization and comparative understanding, Figure 1a depicts a length-scale comparison of different biostructures and micro-fabrication structures [50]. With recent technological developments, it has become commonplace to create devices down to the nanometer level, and of course, microfluidic devices are not an exception to that. The downsizing of microfluidic devices has facilitated the fabrication of nanofluidic devices that usually have at least one characteristic dimension of 100 nm or less [51,52]. The advent of microfluidic and nanofluidic devices (sometimes they are referred to interchangeably) is probably a blessing in life science, analytical biology, and related fields, and they are a good fit for the study of biomolecules as they provide the native aqueous environment with added fluid handling and manipulation capabilities to analyze biomolecules, in some cases even down to the SM levels [53,54,55,56,57]. Usually, SM studies demand a high spatial resolution and often require excitation and probing at the femtoliter-scale volume, which is attainable in microfluidic devices [7,54]. The birth of microfluidics was probably seeded when researchers tried to merge bioanalytical systems with microelectronic disciplines [50,54]. Although some preliminary works were reported [58], the first silicon-based analysis system was developed by Terry et al. in 1979 [59]. They fabricated their gas chromatography air analyzer following standard silicon fabrication methods and were able to reduce the device size by three orders of magnitude compared to the conventional methods at that time. Figure 1b illustrates a photograph of their gas chromatography system with the spiral capillary, input, and exhaust for the gas sample [59]. They have demonstrated the effective use of photolithography, chemical etching, and other fabrication steps necessary to miniaturize a bulkier system; it is thus widely regarded as the first “Lab on a Chip” system and the paper even sometimes referred to as the one that truly gave birth to the field “microfluidics” [50]. Their breakthrough work sparked the growth of microfluidics, and the last three decades have seen an explosive growth of microfluidics, with countless applications in numerous fields [60,61]. The early microfluidic works were mostly focused on microfluidic valves, pumps, inkjet printing, etc. However, the field got a boost during the 1990s, when the researchers in chemical analysis started introducing the breakthrough concepts of “Lab-on-a-Chip” and micro-total analytical systems (μTAS) [62,63,64]. Microfluidics got a bigger acceleration when the Defense Advanced Research Project Agency (DARPA) launched several large-scale projects that were mainly aimed at developing small and portable devices to monitor the environmental and personal health of US military personnel [62]. The Human Genome Project (HGP), which aimed to determine the complete sequence of DNA bases in the human genome, gave another boost to this field during the 1990s [62,65,66,67]. The impetus of these projects, along with the goads of the fundamental and life science fields, sparked incredible research activity in the field of microfluidics and have taken this field to a completely new level, especially for bioanalysis. Figure 1c depicts the chronological progress of microfluidics with notable advancements in the respective timeline [62]. Apart from the silicon and glass microfluidic devices, recently, some alternative materials such as thermoplastics, elastomers, etc. are gaining significant attention. Compared to silicon fabrication, these materials are easy to process, less dependent on cleanroom facilities, and most importantly, are inexpensive and allow for fast replication [68,69]. Towards the end of the 20th century, J C McDonald et al. pioneered the fabrication of microfluidic devices in Polydimethylsiloxane (PDMS) using soft lithographic techniques that highly simplified the device fabrication and started a new era of PDMS-based microfluidic devices [70]. PDMS is so convenient that it almost brought down the device fabrication under the benchtop condition for scientists and engineers, and as should come as no surprise, it became a very popular and handy material for microfluidic devices. PDMS-based microfluidic devices are themselves a vast field with a variety of applications in numerous fields including bioanalysis with a long successful history [68,69,71,72]. Apart from those, 3D-printed microfluidic devices have gained significant attention since the beginning of the 21st century and are employed for different applications including bioanalysis [73,74,75]. Nonetheless, as SM studies deal with individual molecules, it is desirable to access smaller quantities of liquid, preferably in the range of femto/pico litters, thereby requiring the downsizing of microfluidic devices to fulfil the requirements [76]. Apart from ultra-small sample volume requirements, SM studies mostly require low laminar flows, reagent mixing capabilities, integrated fluid-handling capabilities, etc. A number of approaches have already been developed to address these capabilities such as laminar flow cells, droplet microfluidics, multilayer integrated microfluidic devices, etc. Approaches such as 2D and 3D hydrodynamic focusing are sometimes used to mix reagents with reduced mixing times (in the range of µs) while maintaining the laminar flow [77]. Droplet microfluidics is another popular approach for SM studies that enables high-throughput independent and isolated reactions within a small confined volume, which is a desirable feature for SM studies [78,79]. Droplet microfluidics have been used in a number of SM studies, including for DNAs [80,81,82], RNAs [83], direct counting of single molecules [83], amplified detection of DNAs [84], etc. The multilayer microfluidic architecture with valve arrangements allows for precise and controlled fluidic routing and manipulation capabilities with large-scale integration and parallel processing capabilities that facilitates more complex and complete SM studies. Several studies of bioreactors [85], automated mixing [85], pumps [86], DNA analysis [87], etc., are reported on these devices and are getting more attention due to their added capabilities. For further details, the reader is referred to several independent reviews on SM studies using microfluidic devices that can be found elsewhere [88,89,90,91,92,93,94]. Overall, microfluidic devices, along with micro-electromechanical systems (MEMS) and µTAS, have impactful applications in biomedical and SM studies of DNA analysis, immunoassays, cell-based assays, and so on [63,64,68,95,96,97,98].

Most SM studies usually require optical and/or electrical methods along with microfluidic devices [1,7,14,99,100]. Optical integrations are usually carried out using the top-down, confocal microscopic technique, and other optical techniques, which in most cases pose complex and nonplanar integration [99,101,102,103]. Therefore, it is beneficial to develop a platform that can integrate both optics and microfluidics in a single platform, which is referred to as an optofluidic platform. At the beginning of the 21st century, optofluidic platforms have shed new light on SM research as a new horizon with unprecedented capabilities and novel functionalities [104,105,106]. In other words, the optofluidic system intends to incorporate the optical functionalities within the microfluidic chip to avoid the external bulky optics, reduce complexity, lower the cost of the system, and most importantly, miniaturize the system towards a portable size yet with enhanced capabilities. The term optofluidics first appeared in a DARPA-funded research project in 2003 [107]. Since germination, optofluidics has been cordially adopted by researchers from a wide range of disciplines, and thus, with no surprise, it already has established itself as a promising platform. Despite its young age, optofluidics is a burgeoning field that already has provided solutions to practical problems and has demonstrated a variety of important multidisciplinary applications with highly impactful findings and implications for future developments. Optofluidics allows for the integration of rich and versatile optical methods with microfluidics in a single platform that allows for bioanalysis in ideal fluidic settings, with exciting developments at the intersection of photonics, microfluidics, and the life sciences [104,106,108,109,110]. The interaction of light and fluids has opened up avenues for a breadth of optofluidic applications including but not limited to bio-imaging [108], adaptive optical lenses [111,112], optofluidic microscopy [113], lasers [114], biological and chemical sensing [104,115,116], energy harvesting [117], and particle manipulation [110,118]. Figure 2a shows a year-wise publication in the field of optofluidics with a monotonic increment every year [119]. Nonetheless, as optofluidics deals with both light and liquid at small scales, it is imperative to devise a methodology to confine and guide light through the desired part of the fluidic section. Total internal reflection (TIR)-based waveguiding (sometimes referred to as index guiding) is probably the most dominating and vastly used optical method in optical devices [120]. Figure 2b(i) depicts the side view of a typical slab waveguide that guides light based on the principle of TIR [121]. The core of the waveguide (dark gray) has a higher refractive index n_C_, which is surrounded by low-index (n_S_) cladding (light gray). When light is launched in the waveguide at an angle above the critical angle (θ), light is confined within the core following the principle of TIR and cannot escape to the cladding. In optofluidic devices, it is mostly desirable to guide light through the aqueous solutions that usually have a refractive index of 1.33 (approx.), whereas the conventional solid materials that form the liquid core waveguide usually have higher refractive indices (1.4–3.5) [121]. This violates the sole condition of TIR-based guiding and is not suitable for optofluidic devices. Therefore, it is imperative to devise practical methods of light-fluidic interaction for optofluidic devices. To overcome this issue, several clever approaches have been demonstrated, among which some of the approaches are depicted in Figure 2b(ii–vi) [121]. Figure 2b(ii) shows the cross-section of a typical solid core ridge waveguide. Although light is guided within the core following the TIR principle, an evanescent field exists outside the waveguide that decays exponentially [122]. A representative evanescent field is shown with a light gray cross-hatched region that usually exists within a range of some hundreds of nanometers where the interaction of light and liquid takes place. Several studies have been reported on the sensitive detection and trapping of bioparticles and cells that rely on the concept of evanescent field-based guiding [123,124]. Although this concept is well suited for particles that are within immediate contact of the waveguide, this methodology is, however, not ideal for freely diffusing particles that require a comparatively larger interactive volume. There have been attempts to fabricate the liquid core (LC) waveguides with materials that have a refractive index lower than the aqueous solution such as Teflon AF, which has a refractive index of 1.29. As the TIR condition is fulfilled with this approach, it allows index guiding through the LC channel when filled with a suitable solution. Figure 2b(iii) shows a typical cross-section of an LC waveguide with Teflon AF cladding. Several fabrication approaches have been reported based on Teflon AF LC waveguides; however, it remains a challenge to fabricate LC waveguides that can support only a single mode due to the difficulties in controlling the Teflon-AF-based waveguide sidewalls. Another clever approach has been suggested by researchers that relies on reducing the effective refractive index of a high-index material by creating small holes in it, which is referred to as nanoporous cladding. This is a promising approach for optofluidic waveguides which allows for tuning the refractive index (n_NP_) of the cladding material upon necessity [104,125]. A typical schematic of a nanoporous slab waveguide is depicted in Figure 2b(iv). Usually, the nanoporous waveguides provide mode confinement in only one dimension and lack lateral confinement. Liquid–liquid core or so-called L^2^ waveguides are another alternative way to confine light within the desired part of the liquids for optofluidic applications. As the name implies, both the core and cladding parts of these waveguides are formed by two different types of liquids, as illustrated in Figure 2b(v). Wolfe et al. first demonstrated the principle of L^2^ waveguide using CaCl_2_ and water as the core-cladding liquids [125]. L^2^ waveguides provide dynamic tunability and therefore are being used in a variety of applications [126,127]. Slot waveguides are another class of optofluidic waveguides that confine part of the light within a subwavelength dimension core. Slot waveguides are engineered in a way that confines light in a low-index material which is usually sandwiched between two high-index waveguides, as depicted in Figure 2b(vi). To avail this functionality, the width of the low-indexed core is intentionally made smaller than the penetration depth of the evanescent wave in the core medium; thereby, the core width usually falls in the range of hundreds of nanometers. Another notable feature of slot waveguides is the structure is designed in a way that results in a discontinuity of the electric field at a normal boundary between two materials. The structural configuration allows for a greater portion of (usually around 30%) light to be confined within the low-indexed core medium [128]. Although substantial efforts are reported to avail light interaction for studying biomolecules, a smart and efficient way of making an optofluidic platform is based on the antiresonant reflecting optical waveguides usually referred to as “ARROWs”. ARROWs utilize an effective method that uses periodic structures to create a Fabry–Perot etalon that allows light to propagate through the low-indexed liquid medium [128]. A typical ARRROW device developed by Schmidt lab is shown in Figure 2c, where the inset is depicting the generic periodic structure [129,130]. The device has a liquid core waveguide that is surrounded by alternating layers of different refractive indices (*n*_1_ and *n*_2_) and thicknesses (*t*_1_ and *t*_2_). The refractive indices and thicknesses are chosen in such a way that fulfills the anti-resonant condition, where the thickness of the cladding layers can be expressed as follows:(1)$$t_{i}=\frac{\left(2N-1\right)\lambda }{4n_{i}\sqrt{1-\frac{n_{c}^{2}}{n_{i}^{2}}+\frac{\lambda^{2}}{4n_{i}^{2}d_{c}^{2}}}}\;N=1,2,3\ldots .$$
where *n_i_* is the refractive index of the *i*th cladding layer, *λ* is the design wavelength, and *n_c_* and *d_c_* are the refractive index and thickness of the LC channel, respectively. Due to the fulfillment of the anti-resonant condition, light is reflected back to the core medium from the alternating layers, thus enabling low-loss light propagation within the low-indexed LC channel. The device has a long history of successful applications in biomolecule and SM studies, some of which will be discussed in the later sections of this article.

## 3. Single-Molecule Experiments

SM techniques are usually classified into two broad categories: one is the imaging of single molecules, and the other is the manipulation of single molecules [131]. A number of modalities are commonly employed for SM studies, among which the optical and electrical methods have gained particular interest due to their touch-free and non-invasive nature [14,99,132]. SM methods that rely on exerting force on the object of interest such as atomic force microscopy (AFM), optical tweezers, and magnetic tweezers have significant importance in SM studies as they are very handy tools for manipulating individual molecules. These methods are elaborately discussed in the subsequent sections. It probably will not be exaggerating to say that optical methods are the most dominating methods for SM studies, and with no surprise, a pool of optical methods exists for SM studies. Different optical properties are exploited to develop techniques such as fluorescence, absorbance, scattering, optical force, plasmonics, etc. [99,133,134], among which the fluorescence- and optical-force-based methods are given particular emphasis in this review. Fluorescence- and optical-force-based methods of SM studies with some chip-scale applications are discussed in the subsequent sections. Apart from these, electrical methods of SM studies have emerged as incredibly powerful tools for SM studies such as nanopores, nanowire platforms, molecular junctions, carbon-nanotube-based platforms, field-effect transistor based platforms, etc. Among them, nanopores are the most widely used SM tool and remain an active research field with extensive results reported regularly. The last section of this review discusses the electrical methods, especially the nanopore sensors and their application in integrated platforms. Nonetheless, imaging individual molecules is probably the first conceptual and straightforward way of analyzing them. However, due to their incredibly small size, conventional optical imaging is not suitable for imaging single molecules. The first imaging of single DNA and protein molecules was done back in 1956 using an electron microscope [135]. Although it was a big step forward toward SM imaging, it does not ensure the native aqueous environment for biomolecules. The most significant breakthrough in SM imaging probably came along with the fluorescence-based imaging of biomolecules [31,102,136,137]. Fluorescence imaging is a process where the biomolecules are tagged with specific fluorophores that emit light when excited with an excitation light (usually a LASER) of proper wavelength. The wavelength of the emitted fluorescence light is red shifted from the excitation light, thus separation from the excitation light and extraction of the fluorescence light allows for the detection and imaging of single biomolecules. Fluorescence imaging holds the basis of numerous SM studies and is still being used as a handy tool for the detection and imaging of biomolecules. The conventional optical imaging methods are mostly diffraction-limited, thus imaging biomolecules with a high resolution imposes a significant challenge. With cutting-edge light microscopy and improved methodologies, researchers were able to overcome the limitation and find out ways to image single molecules with a high resolution, usually referred to as super-resolution imaging. A number of super-resolution imaging methods have been developed by researchers, including near-field scanning optical microscopy (NSOM) [138,139], photoactivated localization microscopy (PALM) [140], stimulated emission depletion (STED) [141,142], stochastic optical reconstruction microscopy (STORM) [143,144,145], etc., that allow optical imaging down to the single-molecule level. Super-resolution imaging is a huge breakthrough in SM imaging, and as a recognition, the 2014 Nobel prize was awarded to Eric Betzig, Stefan Hell, and William Moerner for their significant contribution to super-resolution imaging [146]. This review is mainly designed to focus on different SM manipulation techniques as described above, and the following sections describe the manipulation methodologies with their functionalities and potential applications. For further details on SM imaging, the reader is referred to other independent reviews on this topic [31,147,148].

### 3.1. Force-Based SM Studies

#### 3.1.1. Atomic Force Microscopy (AFM)

AFM is a powerful tool that is vastly used to image micro/nanostructured surfaces. Usually, the AFM has a cantilever structure with a very sharp tip at the end of the cantilever, and the whole assembly is sometimes referred to as a probe. Typically, the tip has a diameter in the range of a few nanometers, which is swept across the surface to be imaged. AFM works following a similar principle to a scanning tunneling microscope (STM); in fact, sometimes AFM is referred to as a version of STM. The sharp and flexible tip is brought to the proximity of the surface and scanned across it, where the surface roughness is interpreted into the movement of the tip up to the precision of a sub-nanometer resolution [7,149]. The principle of AFM imaging is illustrated in Figure 3a(i) [150] with the detection laser and position-sensitive photodetector [151]. As the AFM probe moves across the imaging surface, there is a net displacement of the AFM tip due to the interaction with the imaging surface. A laser light is shined onto the AFM probe, and the light reflected off the cantilever is monitored via a position-sensitive photodetector. A characteristic mapping of the surface is achieved by decoding the movement of the probe using proper calibration and feedback circuitry [149]. AFM mainly works in three different modes, namely, the contact mode, tapping mode, and jumping mode [152]. Although AFM is primarily used as an imaging tool, it also has a wide application in force measurement and manipulation techniques [149]. The imaging, force measurement, and manipulation capabilities altogether made AFM an ideal tool for bio applications. The extraordinarily high spatial resolution of AFM and accessibility of probing biomacromolecules under physiological conditions have well established AFM as a promising imaging tool for single biomacromolecules [153,154,155]. AFM has been extensively used to image DNA, which is probably the most important and studied biomolecule [156,157,158]. Recently, Pyne et al. have reported a study of the secondary structure and dynamics of supercoiled minicircle DNA using high-resolution AFM [159]. With combined high-resolution AFM and MD simulation, the authors were able to observe intramolecular variations in groove depths and individual DNA defects, and were able to perform measurements on DNA twists. Figure 3a(ii) depicts an image of the supercoiled DNA taken using the high-resolution AFM. Apart from that, a number of cases are reported on DNA studies using AFM such as periodicity measurements of DNA grooves [159], observing uncommon DNA configurations [156,160,161,162,163,164,165,166,167], DNA knots [168], etc. Along with DNA, other biomolecules are also being imaged using AFM such as single proteins [169,170]. Additionally, AFMs have been used in numerous SM studies including of DNA–protein interactions [171,172,173,174], observations of dynamic interactions of transcription factors [175,176], studying protein complexes, protein folding and unfolding [168], etc. Remarkably AFMs are widely used as a prominent tool for force spectroscopy of single molecules. In usual force-based studies, one end of the polymer is attached to the substrate, whereas the other end is weakly coupled with the AFM tip, and thus the polymer forms a bridge between the substrate and AFM tip [177]. As the AFM tip is precisely moved, the polymer is stretched and exerts force on the cantilever, thereby bending it as per the magnitude of the force. Therefore, a thorough analysis of the electrical signal collected from the AFM photodetectors reveals the detail of the force and displacement information. A number of studies are performed for force measurements on DNAs and RNAs [178,179,180], unzip DNA, and RNA molecules using AFM, thereby making it one of the important tools for SM studies [178,181,182]. Although AFM developed itself as a great tool, it has some shortcomings such as unwanted interaction between the tip and imaging surface, nonspecificity, unwanted attachment of molecules, etc. [27]. Nonetheless, AFM remains a great tool for SM studies and has a growing interest and number of applications. For further details on AFM studies, the reader is referred to several independent reviews on AFM-based SM studies [177,183,184,185].

#### 3.1.2. Optical Tweezers

Optical tweezers are one of the vastly used SM methods which utilize optical force to hold and study single molecules. Light has momentum which is proportional to its energy, and a momentum transfer occurs whenever light interacts with objects. If the object is small enough, it is possible to exert sufficient enough force to move and manipulate the object of interest [186,187,188,189]. The principle of the optical tweezer is illustrated in Figure 3b(i) [190]. Generally, lights from a laser source have a Gaussian intensity profile, and whenever it interacts with an object, the forces are scaled in a fashion that pushes the object at the center of the Gaussian beam, provided that the refractive index of the object is greater than the surrounding. If the laser power is high enough, small micro/nanoscale objects and biomolecules can be trapped at the center of the light beam. Optical force-based manipulation was pioneered by Arthur Ashkin starting back in the 1970s [186,191,192], and the first demonstration of a single-beam optical trap, or so-called optical tweezer, was demonstrated in 1986 [188]. The biggest advantage of optical tweezers is probably their touch-free nature of manipulation; therefore, they are highly suited for the study and manipulation of biomolecules [32,193,194]. Ashkin et al. have demonstrated the optical-tweezer-based trapping and manipulation of bacteria, red blood cells, as well as organelles inside cells [187], which probably paved the way for optical-force-based manipulation of biomolecules. A number of studies have been performed to study and manipulate DNA, RNA, protein folding [32,149,195,196,197], bacteria [198], cells [187], etc., using optical tweezers. Figure 3b(ii) shows a typical application of optical tweezers where a confocal microscopic study is performed on DNA molecules [199]. As shown in the figure, the DNA molecule is tethered between two individual optically trapped beads, and a third tightly focused beam (the middle beam, green in color) is used for the fluorescence study of the DNA molecule following the standard confocal microscopic procedure. Although optical tweezers have several advantages, the optical trapping force does not scale well for particles with a smaller diameter; therefore, high optical power is required to trap smaller biomolecules, which has a possibility of damaging the object of interest [194,200]. The phenomenon of damaging objects while studying them using an optical tweezer was reported back in 1986 by Ashkin et al. [188]. Smaller particles (usually in the Rayleigh particles) are more prone to optical damage as higher optical power is required to trap nanoscale particles to overcome the thermal forces [201]. A number of reasons are considered to be responsible for the damage such as photodamage, thermal damage, mechanical damage, etc. [193,202]. Photodamage usually happens due to the linear and nonlinear absorption of light by the object. Biomolecules generally tend to absorb lights in the visible spectrum, and therefore researchers usually use infrared lights to study biomolecules using optical tweezers [193]. Another big concern of optical tweezers is the increase in local temperature around the optical trap, therefore leading to thermal damage of the object [187]. The increase in temperature in an optical trap (ΔT) can go up to several centigrade with an increase in the optical power of 100 mW [187,203,204,205,206]. Biomolecules are significantly affected by the increase in temperature in many ways including molecular stability, functioning, structural stability, etc., and are damaged with a sufficient increase in temperature [193,207]. Therefore, it is undeniable that there is a potential risk of doing lethal damage to biomolecules while studying with optical tweezers [193,202]. Moreover, most optical tweezer setups are non-planar, which sometimes is not feasible for designing compact lab-on-a-chip devices. Nonetheless, as optical tweezers are touch-free, have a high spatial and temporal resolution, and are versatile, therefore, optical tweezers are considered as one of the dominant methods for SM applications.

#### 3.1.3. Magnetic Tweezers

Magnetic tweezers rely on magnetic-force-based spectroscopy, which is developing at a rapid rate and finding applications in different biological studies. The application of the magnetic tweezer was first demonstrated by Strick et al. in 1996 [208], and since then they have been deployed in a variety of applications with necessary modifications for enhanced capabilities [209,210]. The concept of magnetic tweezers is quite straightforward: a magnetic particle experiences a force when placed within a magnetic field which is proportional to the gradient of the magnetic field [210]. Several experimental configurations of magnetic tweezers are available and can be configured based on the experimental requirements. A typical magnetic tweezer setup is depicted in Figure 3c(i), where a pair of magnets (usually permanent magnet or electromagnet) are placed above the flow cell of an inverted microscope setup [211]. The induced magnetic moments exert a net force on the paramagnetic bead that carries the molecule of interest (e.g., DNAs). Generally, the force is manipulated in a way that stretches or twists the molecule. Several studies are reported to exert high forces with relatively small magnetic strength by generating a steep magnetic field gradient using sharp electromagnetic tips [212], small permanent magnets [213], etc. Studying the mechanical properties of different structural forms of nucleic acids is of greater interest in biology, and magnetic tweezers are an ideal tool for probing and taking real-time measurements of the mechanical properties [214]. Figure 3c(ii) depicts an experimental setup where the authors used a magnetic tweezer to analyze force-induced DNA hairpin unfolding [215]. A variety of studies are reported that utilize magnetic tweezers to study and manipulate DNAs [216,217], proteins [218,219,220,221,222], cells [223,224,225,226], etc. Although magnetic tweezers are a great tool for SM studies, the integration of electromagnets is costly and limits the manipulation ability of other techniques [210]. Magnetic tweezers are not as versatile as AFM and optical tweezers, and magnetic tweezers are also limited at weak forces [27]. Nonetheless, the magnetic tweezer is a straightforward tool, and its excellent ability for probing mechanical properties, and especially its unique ability to apply torque and twist biomolecules, has established itself as a widely used tool for SM studies.

**Figure 3 micromachines-13-00968-f003:**
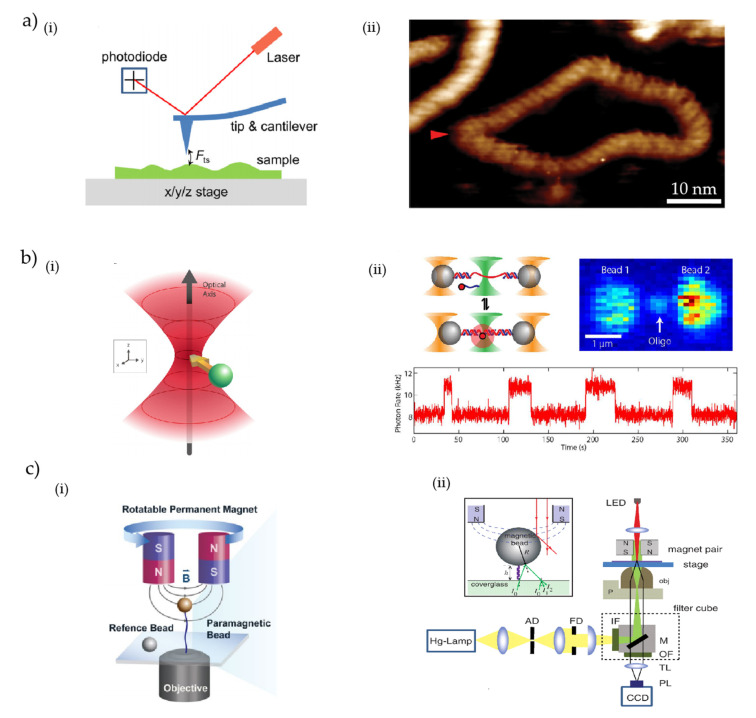
(**a**) (**i**) Basic working principle of AFM. Schematic of typical AFM. The tip-sample force Fts induces a deflection of the cantilever, which is detected by recording the position of the reflected laser beam with a position-sensitive photodiode [150]. (**ii**) High-resolution AFM images of natively supercoiled DNA minicircles [159]. (**b**) (**i**) Schematic representation of the principle of optical tweezer [190]. (**ii**) Combined high-resolution optical tweezers and confocal microscope. Dual optical traps (outer cones) hold polystyrene microspheres (spheres) tethered by a DNA construct (here a DNA hairpin), while a confocal microscope (middle cone) detects fluorescence from a single molecule [199]. (**c**) (**i**) Illustration of the magnetic tweezers setup [211]. (**ii**) Experimental setup for biomolecular calibration based on DNA hairpin unfolding using magnetic tweezers [215].

### 3.2. Fluorescence-Based Single-Molecule Studies

Fluorescence is a technique that relies on the absorption of excitation light by an object (usually fluorophores) followed by a subsequent emission of light at a higher wavelength than that of the excitation light [227,228]. Fluorophores are microscopic molecules (chemical compounds, synthetic polymers, proteins, etc.) that have the fluorescence properties of absorbing light at a shorter wavelength (higher energy) and emitting light at a higher wavelength (lower energy). There are several standard ways of tagging the fluorophores with the biological molecules of interest, a procedure usually referred to as fluorescent labeling [229,230,231]. The absorption and emission wavelengths of fluorophores depend on the outermost electron orbitals in the fluorophore molecule. The wavelength of the emitted light differs from the excitation light by virtue of the electron transition in different energy states generally, referred to as the Stokes shift. The Stokes shift can be visualized and well understood using the Jablonski diagram, which is illustrated in Figure 4a [232]. Without the excitation light, electrons usually reside in the ground energy state, denoted as S_0_ in the figure. When fluorophores are excited with a light of energy *E* ($E=\frac{hc}{\lambda }$, where *h* is the Plank’s constant, *c* is the speed of light in vacuum and *λ* is the wavelength of the light) which is higher than the energy gap between the S_0_ and the lowest excited energy state of S_1_, the fluorophores absorb the photon and a subsequent electron transition happens from S_0_ to S_1_. If the energy of the excitation photons is high enough, then the electron transition can happen in the higher energy states of S_1_ or even S_2_. The electrons eventually come back to the ground energy state S_0_ for energy minimization and stability. While coming back to the lower energy states from the higher energy states, the electrons emit low energy light, as part of the energy is lost due to the vibrational relaxation [233]. Using proper optical filters, it is possible to collect only the emitted fluorescence while suppressing the excitation light and background. The key advantages of the fluorescence methods are specificity and sensitivity. Recent technological advancements pushed the sensitivity to the level where it is possible to collect fluorescence from a single fluorophore [234,235]. There are myriad examples of the use of fluorescence methods that have been reported by researchers around the world [236,237,238,239,240,241].

A few popular fluorescence-based SM experimental methods are depicted in Figure 4b–d [233]. The concept of confocal fluorescence microscopy is illustrated in Figure 4b. As the name implies, confocal microscopy relies on the principle that the excitation and detection optics are focused on the same diffraction-limited spot using pinholes [242]. The main objective of confocal microscopy is to produce a point source to eliminate as much as the stray signals coming from the out-of-focus lights, enabling deep and high-resolution imaging [243] To produce the complete image, the spot is scanned over the imaging sample, data are collected in each spot, and an image is reconstructed from the stack of images [243,244]. As the out-of-focus lights are removed in confocal microscopy, it is possible to achieve high-resolution images with a highly improved signal-to-noise ratio [245,246]. In addition, confocal microscopy has been used for live imaging as well as for fixed samples [243,247,248]. Due to the enhanced capability and high-resolution imaging, confocal microscopy is considered a versatile and effective mode for single-molecule experiments with a growing interest. Figure 4c depicts another elegant fluorescence-based single-molecule method called total internal reflection fluorescence (TIRF) microscopy, which is sometimes also referred to as evanescent wave microscopy [233]. When light travels from a medium of a high refractive index towards a medium of a lower refractive index, the light rays are bent following the principle of refraction. When light incidents are at an angle higher than the critical angle, due to the course of refraction, a portion of the light comes back to the high refractive index medium by a phenomenon called the total internal reflection (TIR) [249,250]. At the occurrence of TIR, light is reflected back to the high-index medium and no longer passes the low-index medium; however, it generates an electromagnetic field that penetrates into the low refractive medium. The electromagnetic field is referred to as the evanescent field that decays exponentially with the depth of penetration, which further depends on the angle of incident, the wavelength of light, and refractive index differences [251]. TIRF microscopy exploits this evanescent field to selectively excite fluorophores at the boundary of the two media without exciting the fluorescence from regions farther away from the surface [252]. As illustrated in the figure, the TIRF is usually implemented on a glass coverslip/slide and physiological buffer interface, where the evanescent field typically remains within a range of 100 nm [233]. A fluorescence excitation using the evanescent field within that very thin zone enables imaging with a very low background and minimal light exposure [250,253]. Therefore, TIRF enables better contrast as it collects minimum out-of-focus light and significantly improves the signal-to-noise ratio, making itself an ideal tool for probing single molecules [233,245,246]. As TIRF permits visualization of biological events on an unprecedented scale, it is probably one of the most used fluorescence methods for single-molecule experiments, with numerous findings already reported and more to come in the near future [233]. Figure 4d shows another powerful tool of fluorescence-based single-molecule experiments, namely the Förster Resonance Energy Transfer (FRET) [233,254,255,256]. FRET is a process of nonradiative energy transfer first observed and quantitatively described by Theodor Förster in the 1940s [257,258]. The energy transfer in FRET takes place by a nonradiative dipole–dipole coupling where the energy is transferred from an excited fluorophore (donor) to a nearby chromophore (acceptor) [259]. Usually, FRET is observed within a distance of 10 nm (donor–acceptor separation distance), making it a unique method for studying and retrieving information on molecular proximity, orientation, and conformational dynamics, with a resolution far below the diffraction limit of optical microscopy [259,260]. As FRET is highly sensitive to intermolecular distance and orientation, it is extensively used to study molecular interactions such as protein–protein interactions and protein–nucleic acid interactions, as well as protein dynamics, the cytoplasm, and so on [233]. FRET is also combined with other SM methods and is revealing exciting findings, with a growing interest in a variety of applications.

### 3.3. On-Chip Optical Detection of Single Molecules

There is an increasing demand for developing on-chip sensing platforms of biomolecules to meet the recent trends in personalized healthcare, rapid point-of-care diagnostics, etc., leading on-chip sensing as an active and expanding field of research [261]. A myriad of on-chip sensing techniques have already been developed based on different methodologies including optical, plasmonic, and electrical modalities, some of which have yet to develop into full-fledged commercial healthcare applications [262,263,264,265,266,267]. Recently, plasmonic devices have been getting particular attention in biosensing applications. Plasmonics deals with surface plasmon resonance (SPR), which can dramatically increase the light confinement and near-field enhancement at the nanoscale, are thus capable of boosting the analyte sensitivity [261]. Recently, Guardado et al. reported the detection of NS1 protein biomarkers related to the Dengue virus developed on a plasmonic optofluidic device [268]. The schematic and principle of their methodology are depicted in Figure 5a. The device can perform the pre-treatment and separation of plasma directly from a blood sample and the subsequent detection of NS1 biomarkers. To avail the functionality, the authors have integrated a cavity-coupled plasmonic biosensor integrated with a microfluidic blood plasma separator, as illustrated in Figure 5a. The plasmonic nanostructure used in this work was a combination of localized surface plasmon resonance and an asymmetric Fabry−Perot cavity resonator. Using their methodology, the authors were able to demonstrate the detection of NS1 biomarkers in a clinically relevant range at a concentration of 0.1 to 10 μg/mL^−1^ in bovine blood.

Liu et al. have developed an integrated biosensing tool based on a slot waveguide Mach–Zehnder interferometer (MZI) that is capable of real-time, label-free detection of microRNAs (miRNAs) in clinical urine samples [269]. The schematic of their miRNA detection methodology with different sections is shown in Figure 5b. The miRNA detection principle mainly relies on the measurement of the change in the phase caused by the hybridization of the complementary DNA capture probe and the target miRNA. The device was fabricated following the standard Si fabrication technique, and a laser light of 1562 nm was used for optical measurements. The formation of the double-stranded structure between the complementary DNA capture probe and the target miRNA induces a phase change between the two arms of the MZI, which was extracted from the output intensity variation. The authors have demonstrated the real-time, label-free detection of miRNAs using their device. The device provides good enough sensitivity to be used in the clinical range (≤1 fmol μL^−1^), and most importantly, offers specificity and does not require a pre-incubation step or temperature control. Therefore, it has the potential to be used as a point-of-care diagnostic tool.

Apart from these biomolecule sensing platforms, waveguide-based optofluidic devices have recently gained special attention as they offer planar, highly functional, and reconfigurable devices with further integration capabilities [106,116,270,271,272,273,274]. The ARROW optofluidic platform pioneered by the Schmidt group has recently been used in a wide range of single-molecule applications for a variety of biomolecules. A schematic representation of the generic ARROW optofluidic device is depicted in Figure 5c. The devices are fabricated following standard silicon fabrication techniques with alternating layers of SiO_2_ and Ta_2_O_5_ with suitable thicknesses to fulfill the anti-resonant condition. As shown in the figure, typical ARROW devices have a “Z” shaped LC channel (blue) and intersecting solid core (SC) waveguides (gray). The SC waveguides are used to introduce and collect light to and from the device, whereas the LC channel is used for both optical and fluidic handling purposes. Reservoirs are glued at the end of the LC channels, which are used to introduce analyte samples into the LC channel. By adjusting the fluidic levels on the reservoirs, it is possible to control the fluidic flow inside the LC channels, and the electrical potential can be applied to the analytes via suitable electrodes. Usually, optically labeled analytes are introduced in one of the reservoirs (inlet), and a fluidic flow is maintained from the inlet to the outlet reservoir maintaining a pressure-driven flow. Fibercoupled laser lights with suitable wavelengths are coupled into the excitation SC waveguide (red arrow) which is guided to the LC channel excitation region. As the analytes pass through the excitation region, it is excited by the laser light and emits fluorescent light. A part of the fluorescence light is collected via the orthogonal LC SC waveguides (green arrow) and sent to a photodetector (after filtering), which produces a detection spike as shown in the inset of Figure 5c. The device has sufficient enough sensitivity to detect nucleic acids [275] and viruses [276,277] and has been employed for a pool of applications such as particle trapping and manipulation [278,279,280], optical filtering [281,282], particle sorting [283], atomic spectroscopy [284,285], surface-enhanced Raman spectroscopy (SERS) detection [286], etc. The group has also demonstrated simultaneous multiplexed detection of single viruses using their ARROW optofluidic devices by taking advantage of the interference within a multimode interferometer waveguide [276]. With a careful design, they used a wider SC waveguide, referred to as the multi-mode interference (MMI) waveguide, that can support multiple modes which intersect with the LC channel as depicted in Figure 5d(i) [276]. Depending on the physician dimension of the waveguide and the wavelength of the excitation light, a spatial optical pattern is generated consisting of multiple peaks in the time domain transforming the spectral information by virtue of the optical interference [276]. The number of optical spots (N) can be related to the excitation wavelength (*λ*) using the following equation:(2)$$N\lambda =\frac{n_{c}w^{2}}{L}$$
where *w* is the effective MMI waveguide width (in this case, 100 μm), *n_c_* is the effective refractive index of the MMI (in this case, 1.46), and *L* is the MMI length (in this case, 3.4 mm) [276]. Figure 5d(ii) illustrates the optical patterns generated within the intersection with the LC channel region when excited with laser lights having different wavelengths [276]. It should be noted that this time, a single wavelength laser light produces multiple bright spots (peaks) within the LC channel, and the number of peaks changes as the wavelength of light changes. For example, excitation with a 488 nm laser light produces nine distinguishable spots, whereas a 745 nm excitation produces six peaks, and accordingly, when properly labeled single particles pass the excitation region, they are supposed to produce nine and six peaks in the fluorescence detection signal, respectively. Therefore, by observing the peak pattern in the detection signal, it is possible to distinguish particles from a mixture, which is the key concept of the multiplexed single-molecule detection methodology. For experimental demonstration, the authors optically labeled inactivated H1N1 and H3N2 influenza viruses with blue and dark red color dyes, respectively, whereas the H2N2 virus was colabeled with both blue and dark red dye. Accordingly, the labeled viruses were excited with the mentioned two colors, and the subsequent fluorescence signal was collected after proper filtering. As depicted in Figure 5e, the nine peak fluorescence signals (top) were produced by the H1N1 viruses, the six peak fluorescence signals (bottom) were produced by the H3N2 viruses, and the H2N2 viruses produced a fluorescence signal that is a mixture of six and nine peaks [276]. Therefore, the methodology allows for the multiplexed detection of single molecules by analyzing the temporal signal using different signal processing methods toward versatile diagnostic instruments for a variety of bioparticles and biomarkers.

**Figure 5 micromachines-13-00968-f005:**
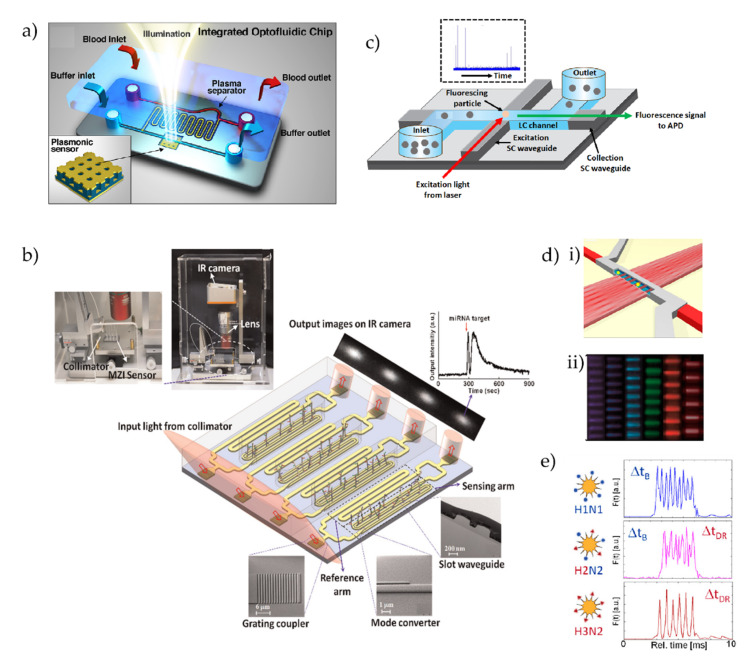
(**a**) Integrated Dengue virus biosensor comprising an in-line microfluidic blood plasma separator and a cavity-coupled nanoimprinted plasmonic array. Inset corresponds to the nanoimprinted plasmonic biosensor composed of a gold mirror (80 nm), a dielectric film spacer (∼700 nm) embossed with a square array of holes, a conformal thin film of aluminum oxide layer (20 nm) as the fluid barrier, and a thin gold film (30 nm) [268]. (**b**) Schematic diagram of the MZI biosensor system for miRNA detection [269]. (**c**) Principle of fluorescence-based particle detection in ARROW optofluidic devices. Inset: an actual APD trace of particle detection (Figure 5c is used upon permission from the Applied Optics group at the University of California, Santa Cruz, CA, USA]. (**d**) (**i**) Schematic view of MMI waveguide intersecting with a fluidic microchannel containing target particles [276]. (**ii**) Photographs of multispot excitation patterns created in fluidic channel filled with fluorescent liquid. The entire visible spectrum is covered by independent channels (405 nm/11 spots, 453/10, 488/9, 553/8, 633/7, 745/6). (The original black and white color scale was rendered in the actual excitation colors) [276]. (**e**) Schematic view of the labeling scheme for the three influenza types and their resulting single-virus fluorescence signals; the H2N2 virus shows a mixture of six and nine peaks upon blue and dark red excitation [276].

### 3.4. Sorting Single Particles

The necessity of separating and sorting cells and biomolecules from a mixture is ubiquitous in biology and the life sciences [287]. Several methodologies exist to sort particles such as sieving, sedimentation, etc. However, for sorting smaller particles, more advanced methods such as optical methods, dielectrophoresis, etc., are more suitable [288]. Optical methods to separate and sort particles are quite popular due to the virtue of their touch-free, noninvasive nature and other obvious advantages [289,290]. As stated earlier, it is possible to exert force on particles using optical beams due to the momentum transfer during the collision of light and particles. Figure 6a shows a smart method of particle sorting that relies on the balance between the fluidic and optical force on particles to selectively remove particles from a channel based on size [189,283,291,292,293]. The sorting method is devised by Kaelyn et al., which was developed on an “H” shaped ARROW optofluidic device as depicted in Figure 6a(i) [283]. The device has interconnected liquid core (LC) channels with four reservoirs at the end of the channels as numbered from 1 to 4. The reservoirs are used to input the analyte particle mixtures and/or balance the fluidic flow across the channels. In the first setting, a water-based solution of a mixture of 0.25 µm, 0.5 µm, 1.0 µm, and 1.5 µm sulfate latex beads was inputted into reservoir 1, whereas the other reservoirs were filled only with the fluidic solution. The fluidic pressure was maintained in a way that the net flow was directed from reservoir 1 to reservoir 2 (along the *Z*-axis). As shown in Figure 6a(i), the device has several solid core (SC) waveguides, shown as the gray ridge in the figure [283]. A fiber-coupled laser light (532 nm) was coupled in one of the SC waveguides (indicated as a green arrow), which is coupled to the LC channel at the “T” intersection as depicted in Figure 6a(ii) [283]. As the particles are moving within the LC channel, they have a fluidic force acting upon them (along the *Z*-axis). In addition, as the laser light is propagating orthogonally to the flow direction (along the *X*-axis), the particles have an optical force acting on them, too. As the optical force is size-dependent, particles with larger diameters experience a higher optical force compared to the particles with smaller diameters [293]. By adjusting the laser power and fluidic flow, it is possible to scale the forces in a way so as to selectively push larger particles out of the flow stream, which is the core principle of particle sorting in this case. As the mixture of particles moves from reservoir 1 to reservoir 2, the larger particles (>1 µm) were pushed towards reservoirs 3 and 4, while the smaller particles were able to continue towards reservoir 2, thus effectively separating particles from a mixture. The authors have shown a second configuration for particle sorting in the same device, which is shown in Figure 6a(iii) [283]. In the second setting, the particle mixtures were loaded in reservoirs 3 and 4, and a resultant flow was maintained towards reservoir 2. In this case, the laser light was coupled to the SC waveguide that propagates light in a direction opposite of the flow direction towards reservoir 2. Again, the laser power and flow velocity were adjusted in a way that pushes the larger particles (>1 µm) towards reservoir 1, eventually separating them from the mixture. Apart from optical methods, dielectrophoresis (DEP) is another remarkable technology that is used in microfluidic and optofluidic devices for separation, trapping, and many other applications [288,294,295,296,297,298,299,300,301]. Figure 6b illustrates the details of a DEP-based particle separation method in a microfluidic device [288]. DEP is a phenomenon where a charge-neutral but polarizable particle experiences a force in a non-uniform electric field due to the interaction of the particle’s dipole and the spatial gradient of the electric field [302,303,304,305]. Jason et al. have demonstrated a methodology of particle separation based on DEP in a microfluidic device whose top and cross-sectional views are shown in Figure 6b(i) and Figure 6b(ii), respectively [288]. In a PDMS device, the authors have fabricated an array of slanted, planar, and interdigitated electrodes, as depicted in Figure 6b(ii) [288]. A photograph of the actual device is shown in Figure 6b(iii) [288]. A mixture of 4-µm and 6-µm particles suspended in an aqueous liquid were loaded at the inlet of the device. As the mixture of particles flow through the microchannel and come across the slanted electrodes, they experience a transverse force due to the DEP phenomenon, which is dependent on the particle size and field strength. The larger particles experience a larger DEP force compared to the smaller ones; therefore, they are more deflected compared to the smaller particles as they flow down the length of the device. The experimental event was recorded with a camera, and a snapshot of the separated microparticles is shown in Figure 6b(iv), where the separation between the 4-µm and 6-µm microparticles is visible [288]. A more complete and meaningful optofluidic setup is shown in Figure 6c that can sort and subsequently count the circulating tumor cells (CTCs) in a single optofluidic platform [306]. CTCs are tumor cells that are removed from primary tumors and mixed with the bloodstream [307,308,309]. Therefore, detecting and monitoring CTCs in peripheral blood has a meaningful impact on cancer diagnostics [310,311,312,313,314]. Qingling et al. have demonstrated an optofluidic flow cytometer (OFCM) that can consecutively separate CTCs followed by a single-cell phenotypic counting. The CTCs were separated based on the physical characteristics of CTCs that are independent of biomarker expression. To avail the CTC separation, the authors have designed a PDMS-based multistage microfluidic chip. The first stage of the chip consists of a separation structure employing spiral channels using inertial focusing. In the second stage, the device had a 3D hydrodynamic focusing structure with a 90° curved channel connected to a straight channel along with two horizontal sheath channels symmetrically distributed on both sides of the straight channel, and a flow resistance matching region using a serpentine channel to ensure the consistency of flow resistance in the two outlets of the first stage [306]. As the ratio of the inertia lift force (F_L_) and Dean drag force (F_D_) is proportional to the particle size, CTCs and some larger white blood cells (WBCs) are focused on the equilibrium position, whereas the red blood cells (RBCs), WBCs, platelets, and plasma proteins are excluded. For improved fluorescence detection, the authors have precisely designed a 3D hydrodynamic focusing structure that keeps the cells at the center of the detection channel. In the following section, the authors have designed a fluorescence detection module to detect fluorescence from the fluorescently labeled CTCs. Laser lights from a 488-nm and 638-nm laser were used for optical excitation of the labeled CTCs, which emits four color fluorescence from individual CTCs. Three dichroic mirrors and subsequent filters were used to separate and collect the green (510–530 nm), yellow (560–590 nm), red (650–690 nm), and near-infrared (700–740 nm) fluorescence. The four separated fluorescence signals were sent to four individual photodetectors. The APD data were further processed using a custom-written MATLAB code which enabled the authors to count the CTCs. Cipriany et al. have demonstrated another sorting platform based on a nanofluidic device that can specifically identify color-coded DNAs and perform sorting based on the identification [315]. The methodology of the sorting principle is depicted in Figure 6d and consists of bifurcated nanofluidic channels to enable different routing for sorting. To detect and identify individual DNAs, the authors used a fluorescence detection scheme. The authors designed a field programmable gate array (FPGA)-based digital signal processing algorithm on the collected fluorescence signal to perform real-time analysis for proper identification of the DNAs. Once the methylated DNAs were identified, a voltage actuated flow was activated to drive the target DNAs towards the desired outlet. In their first step, the authors used a mixture of DNAs having different lengths and were able to perform a size-based identification based on the intensity of the collected fluorescence signal. In their next step, the authors demonstrated a sorting of methylated DNA from a mixture of methylated pML4.2 and unmethylated pUC19 DNAs. A two-color labeling scheme was used to distinguish the methylated and unmethylated DNAs. At first, both of the DNA species were labeled with the red TOTO-3 dye and subsequently mixed with green-labeled MBD1, which specifically binds with methylated DNA targets and represses gene expression [316] The FPGA-based detection system identified the methylated DNAs by analyzing the two-color fluorescence signal and sorted them accordingly based on the voltage-based fluidic routing. Following the methodology, the authors have demonstrated the identification and sorting of methylated DNAs with an efficiency up to 98%. This methodology can be used to sort specific DNAs and can find application in epigenetic analysis studies. Apart from the above-mentioned methods, several other methods for particle sorting exist such as acoustic [317,318], mechanical [319,320,321], thermosensitive hydrogel [322], etc., which are finding applications in a variety of fields including the SM experiments.

### 3.5. Trapping Single-Molecules

As stated earlier, studying and analyzing single molecules is probably the best way to improve our understanding of biomolecules and molecular processes and to use the wealth of unprecedented levels of information in numerous applications across a number of disciplines. For a prolonged study, one would expect to immobilize the molecule during the course of the experiments. However, due to obvious reasons, the biomolecules do not remain standstill in their native aqueous environment [323,324,325,326,327,328,329,330]. Therefore, for a thorough and prolonged study, it is imperative to develop methodologies that can hold the biomolecules in their place or restrict their movement in a highly controlled manner in the molecule’s native environment, which is referred to as trapping. Several methods have been developed for particle trapping including the pioneering optical methods of molecular trapping. Despite being massless, photons can transfer momentum and therefore are capable of exerting force on the objects they collide with. Usually, when a collimated laser beam hits a particle, there are two components of optical force that act on the particle. One component of the optical force that pushes the particle in the direction of propagation is generally referred to as the scattering force. The second component of the optical force usually acts along the intensity gradient of the optical beam, which is typically known as the gradient force. A resultant of the two forces acts as an overall force on the particle, therefore influencing the movement of the particle [303,331,332]. Since the first demonstration of optical manipulation by A. Ashkin in the 1970s, optical trapping and manipulation methods remain the gold standard for probing and trapping biomolecules [186]. Figure 7a shows an optofluidic platform that can attract, trap, and propel particles based on the evanescent field produced at the surface of a waveguide [333]. Bradley et al. have demonstrated an evanescent-field-based trapping of dielectric particles on an optofluidic platform that has SU-8 epoxy-based photonic structures and PDMS microfluidics on a fused silica substrate. The microfluidic channel is fabricated just on top of the optical waveguide in a way that, upon optical excitation, the evanescent field extends towards the fluidic section and exerts an optical force on the particles flowing through the microfluidic channel. During the experiment, dielectric particles were loaded and flown through the microfluidic channel based on a pressure-driven flow. As the particles arrived at the optical excitation region, the evanescent field trapped the particles and pushed them towards the optical propagation direction, which is perpendicular to the pressure-driven flow. As expected, the authors observed that successful trapping depends on the balance of the optical power and the flow velocity of the particles. In successful trapping events, the particles were pushed to the microchannel wall and remained captured as the trapping force was reduced. Francesca et al. have demonstrated a dual-beam optical trap with cell-stretching capabilities implemented on an optofluidic device [333]. A schematic representation of the chip is depicted in Figure 7b [334]. The device was fabricated in a fused silica glass substrate using 3D femtosecond-laser micromachining. As shown in the figure, the device has a fluidic microchannel for fluidic transport, whereas the optical trapping is designed following the principle of dual-beam traps using two counterpropagating optical beams from the perpendicular integrated optical waveguides. The principle of dual-beam optical trapping is quite straightforward. A particle experiences an optical force when hit by a laser beam along the direction of propagation. In a dual-beam trap, two optical beams propagating opposite to each other hit a particle, and the particle experiences counteracting forces on it. By adjusting and equalizing the optical forces, it is possible to capture and trap a particle, which is the underlying principle of trapping in this case. Following the principle, the authors have demonstrated the trapping of red blood cells (RBCs) that are delivered to the trapping region using a controlled flow mechanism within the fluidic channel. In addition to trapping, they have also demonstrated the stretching of RBCs with higher optical power, which has fundamental importance in the cellular system [335,336]. Optical trapping using a photonic resonator structure is another smart way of trapping biomolecules. Sudeep et al. have demonstrated an optofluidic structure that exploits optical resonance in one-dimensional silicon photonic crystals to trap dielectric nanoparticles [337]. A schematic representation of their device is illustrated in Figure 7c(i) [337]. As depicted in the figure, the device consists of a silicon (Si) waveguide named the bus waveguide along with a 1D photonic crystal micro-cavity just adjacent to the Si waveguide. The device has PDMS microfluidic channel orthogonal to the Si waveguide that was carefully aligned with the resonators during bonding. The device is designed in a way that the 1D photonic crystal resonator evanescently couples to a bus waveguide. The resonant optical field within the resonator resembles a standing wave that enables a static trapping point with strong field confinement in all three dimensions. During the experiment, a TE polarized light from a 1548.15-nm laser was coupled into the bus-waveguide, which formed a stationary interference pattern within the photonic crystal resonator that ultimately resulted in a trapping spot by confining the optical field in a tiny volume, as illustrated in Figure 7c(ii) [337]. Using this methodology, the authors have demonstrated the trapping of 48 nm and 62 nm dielectric particles. This method offers better trapping stiffness compared to optical tweezer trapping or slot waveguide trapping, and it is thus a suitable method for single-molecule trapping. Figure 7d depicts another category of optical trapping implemented on an ARROW optofluidic device developed by S. Kühn et al. [279]. The optofluidic chip has intersecting LC (blue) and SC (gray) waveguides as shown in the figure. To enable trapping, two counter-propagating optical beams were launched into the microchannel via the SC waveguides from the left and right sides of the device. The device has another pair of SC waveguides perpendicular to the trapping SC waveguides which can be used for further interrogation of a trapped particle at the intersection, as depicted in the figure. This particular trapping method takes advantage of the waveguide loss to form a dual-beam trap, and it is therefore referred to as a loss-based optical trap. The 1-µm beads were introduced into the LC channel via one of the glued reservoirs at the end of the LC channel. Two optical beams from a 532-nm laser were coupled to the left and right trapping SC waveguides. The two optical beams were propagated from the left and right sides of the LC channel via the SC–LC interfaces, ultimately forming the counter-propagating dual-beam trap. As the optical beams hit a microbead inside the LC channel, by virtue of the optical scattering force, the optical beams push the microbead along their direction of propagation; therefore, the microbead experiences a counteracting optical force from the two beams. By adjusting the power of the left and right optical trapping beams, it is possible to equalize the counteracting optical forces at any point along the LC channel, thus enabling particle trapping at any desired location of the LC channel. Following the loss-based (LB) trapping methodology, the authors have demonstrated the successful trapping of a single microbead. Once the fluorescent microbead was trapped at the intersection, the bead was optically excited using a second laser light (632 nm) introduced via the perpendicular (w.r.t to the trapping beams) SC waveguide. Using this methodology, the authors performed a subsequent fluorescence analysis of the trapped microbead. In addition to that, the authors also showed a successful demonstration of trapping *Escherichiea coli* (*E. coli*) bacteria that were attached to sulfate-terminated polystyrene beads, which in turn demonstrates the true power of optofluidic platforms towards on-chip trapping and prolonged analysis of biomolecules. Optical trapping has a long successful history in manipulating and trapping biomolecules. However, the optical force does not scale well for small particles, especially particles having a diameter of less than 100 nm [278,338]. Optical trapping of smaller particles requires prohibitively high optical power, which is a big concern. To overcome this issue, researchers come up with an elegant approach that relies on optically tracking the particle’s movement from the trapping point and subsequently applying an electrical feedback force that brings the particle back to the point of interest, therefore trapping the particle. With a fast enough feedback arrangement, it is possible to restrict the Brownian motion, and therefore the trap is referred to as the anti-Brownian electrokinetic (ABEL) trap. Cohen et al. first demonstrated the principle of the ABEL trap, and the details of their methodology are depicted in Figure 7e [338]. The trapping device was built in a glass cell for easier fluorescence imaging of the particle of interest. The device had four fluidic channels to facilitate voltage applications from macroscopic control electrodes to the trapping region (Figure 7e(i,ii)] [338]. Using an inverted fluorescence microscope, images of the fluorescence-labeled particle were acquired on a high-sensitivity digital camera with a frame rate of up to 300 Hz. Using custom-coded image processing, the two-dimensional coordinates of the particle’s movement were acquired in real-time on a frame-to-frame basis. Once the particle’s position was measured for a particular frame, a corresponding feedback voltage was applied that was proportional to the offset between the measured position and the desired trapping position. The electrokinetic force due to the voltage application (VαF) pushed the particle back to the desired trapping position before the arrival of the next video image, and the process repeats. Following the principle of ABEL trapping, the authors have demonstrated the trapping of fluorescently labeled tobacco mosaic viruses (TMV; ≈300 nm long × 15 nm in diameter). The trapped trajectories of 13 TMVs are shown in Figure 7e(iii), where it is clearly seen that the viruses are confined within 2 µm, which in turn illustrates the efficacy of the ABEL trap [338]. A remarkable feature of the ABEL trap is that it tracks a particular particle and applies feedback force that only corresponds to that particle and is considered random and uncorrelated for the other particles. This feature allows for trapping only one particle at a time, making the ABEL trap a true single-molecule tapping methodology. The ABEL trap has also been implemented in other optofluidic platforms and with different configurations, and successful trapping of biomolecules are reported [278,339]. The ABEL trap has been successfully implemented to trap a single dye molecule, which demonstrates the true power and unprecedented capability of this tapping method [340].

DEP trapping of biomolecules is another popular and widely used method in SM studies. As stated earlier, DEP is a phenomenon where a polarizable particle can be moved when placed in a non-uniform electric field with an intensity gradient. DEP trapping offers fast, non-invasive, force-based manipulation of biomolecules [342,343]. Ghomian et al. developed a low-cost DEP-based trapping platform that offers easy and reproducible fabrications of devices [341]. The schematics and layout of their device are shown in Figure 7f(i,ii). where a pair of gold nanoelectrodes are used to apply the AC DEP voltage. The device was fabricated using the standard Si fabrication technique. However, the gold nanoelectrodes were fabricated using projection photolithography instead of electron-beam lithography to reduce the device cost. It should be noted that an asymmetry in the width (one was 400 nm and the other was 600 nm) of the electrodes was intentionally maintained in order to enhance the electric field gradient in between the electrodes. An SEM image of the electrodes within the device is shown in Figure 7f(iii). The authors have designed an electronic circuit for the purpose of trapping, monitoring, and better functionality. In their study, the authors have demonstrated the trapping of 1D 10-Helix-Bundle (10 HB) DNA origami for over several minutes. Following the fluorescence microscopic technique, the authors were able to image the trapped DNA, and a representative picture of trapped DNA origami is depicted in Figure 7f(iv) [341]. Additionally, their methodology consists of a capacitance monitoring feature by which the authors were able to observe the DNA origami concentration around the trap location by monitoring the change in capacitance during the trapping process. Apart from the mentioned trapping methodologies, a myriad of other methodologies are developed for trapping biomolecules, and the reader is referred to several independent reviews [29,344,345,346] for further details on this topic.

### 3.6. Electrical Methods for Single-Molecule Experiment

Apart from the optical methods, electrical methods for the detection, control, probing, and manipulation of single molecules is another vast set of experiments that is rapidly expanding with growing interest [14,132,347,348]. Electrical approaches are particularly fascinating as they offer real-time, label-free, high-resolution single-molecule measurements with easier integration capabilities [348]. A number of electrical methods are already developed for single-molecule experiments such as nanopores [34,349,350,351,352,353], nanowire platforms [354,355], molecular junctions [356,357,358], carbon nanotube based platforms [359,360], atomically thin nanoribbon platforms [359,361,362,363], organic field-effect transistors [364,365], etc. With the recent technological advancement and renaissance of nanotechnology, the electrical single-molecule methods have demonstrated their efficacy and excellent potential, and nanopores have especially demonstrated their unprecedented capabilities and placed themselves as a pioneering tool for single-molecule experiments [366,367,368,369]. A nanopore is a nanoscale opening formed by a natural protein or artificially fabricated in a thin insulating membrane [370]. With proper sealing, both sides of the nanopore are filled with ionic liquids, therefore leaving the nanopore as the only contact bridge between the solutions of the two sides [371,372,373]. An application of a voltage across the two sides forms an ionic current through the nanopore, which is usually termed the baseline current. Analyte molecules are introduced on one side of the nanopore, and an application of voltage with proper polarity and magnitude drives the analytes from one side (cis) of the nanopore to the other side (trans) through the nanopore, and this event is usually referred to as translocation. As the analytes pass through the nanopore, they transiently modulate the current, creating a momentary spike in the current signal which is considered a signature of analyte detection [374,375,376]. Usually, the nanopores are sensitive enough to resolve the physical size, structure, charge, and dynamic interaction of analyte molecules which are encoded within the detection signature; therefore, the detection spikes contain a wealth of crucial information about the analyte molecules which can further be extracted upon proper analysis [374,377]. The working principle of the nanopore is illustrated in Figure 8a, with a typical current trace and detection spike [14]. Since the germination and demonstration of nanopores by D. Deamer [378], this technology has created a reverberation and seen explosive growth in the field of DNA sequencing and single-molecule sensing [349,379,380,381]. Nanopores are usually classified into three broad categories. The first category is the one that is found in nature as molecular biostructures, such as α-hemolysin [382], MspA [383], viral connectors such as phi29 [384], etc., that are generally referred to as biological nanopores. The second category is the one that is artificially fabricated on thin solid-state membranes called solid-state (ss) nanopores [370,377]. The third category is the amalgamation of the biological and ss nanopores usually termed as hybrid nanopores [38,385,386,387,388]. The primary focus of nanopore technology is nucleic acid sequencing. If a nanopore is made sensitive enough that it produces distinguishable current modulation as different bases of nucleic acids pass through the pore, it is possible to sequence the nucleic acids by analyzing the nanopore current trace, which is the core idea of nucleic acid sequencing [38,389] The principle of nanopore sequencing is illustrated in Figure 8b [389]. With the extensive efforts of numerous researchers around the world, it was feasible to develop a nucleic acid sequencing tool based on nanopores [380,390,391]. Nanopore-based nucleic acid sequencing is a vast topic, and readers are referred to explore the independent reviews on this topic for further details [17,380]. Realizing the effectiveness of nanopore-based nucleic acid sequencing, Oxford Nanopore Technologies has developed and commercialized its first DNA sequencing prototype, and some other companies are developing nanopore-based sequencers [392,393]. Although nanopores were primarily developed for nucleic acid sequencing, due to their effectiveness and versatility, nanopores are employed in a variety of applications, especially for probing single molecules. A number of applications of nanopores are already reported to detect and probe a variety of biomolecules such as single-stranded (ss) DNA [394,395,396,397,398,399,400], double-stranded (ds) DNA [400,401,402,403,404,405,406], RNA [407,408,409,410], proteins [275,411,412,413,414], ribosomes [275,415] etc. Another remarkable feature of nanopores is their integration capabilities. Nanopores, especially the ss nanopores, allow for easy integration with conventional single molecule tools such as the vast pool of optical methods and microfluidic and optofluidic platforms, making them a general molecular analysis platform for a variety of other applications [14,101]. Recently, the optofluidic integration of nanopores has gained impetus toward developing economic, portable, chip-scale, single-molecule probing and diagnostic tools [416,417,418]. An interesting application of such integration is the simultaneous electro-optical detection of single molecules. The methodology is simple, with an optically labeled biomolecule being first electrically detected using nanopores followed by subsequent optical detection of the same biomolecule [419,420]. As both the electrical and optical detection signature comes from the same molecule, they are correlated, and the methodology increases the confidence of detection, provides more information for further analysis, and has the potential to address challenges in nucleic acid sequencing [277,421]. A number of methods are reported on the electro-optical detection of single molecules including implementation on optofluidic platforms [419,420].

Although nanopores are a great tool for SM experiments, there is still room to improve the performance of the nanopore itself. When a voltage is applied across the nanopore, a major part of the electrical field is confined within a small radius around the nanopore (typically within a few microns), usually referred to as the capture radius [423,424]. Therefore, nanopores mostly capture the analytes available within the very tiny capture volume, which in turn limits the nanopore’s performance [14,425,426,427]. To overcome this limitation, Rahman et al. have demonstrated an elegant solution developed on the previously described ARROW optofluidic platform that can improve the performance of nanopore sensing by increasing the detection rate of analytes [422]. The core idea of their methodology is to accumulate the analytes and release them within the vicinity of the nanopore, aiming to increase the local concentration within the capture volume of the nanopore. The authors used functionalized microbeads to carry the target DNAs (corresponding to a melanoma cancer gene in this case), and the target carrying beads were optically trapped under the nanopore using the principle of the loss-based tap. The principle of the nanopore capture rate enhancement is depicted in Figure 8c(i), which the authors referred to as the trap-assisted capture rate enhancement (TACRE), and the schematic of the whole device is shown in Figure 8c(ii) [422]. A nanopore was drilled on the LC channel of the ARROW device using a dual-beam electron microscope. Accordingly, a total of three reservoirs were glued to the device, among which two were glued on the ends of the LC channel (reservoirs 1 and 3), and reservoir 2 was glued on top of the nanopore. An aliquot of a target-carrying bead sample was loaded on reservoir 1, where the target DNAs were attached with functionalized magnetic beads following standard procedure [422]. The liquid level of reservoirs 1 and 3 was maintained in a way to maintain a controlled pressure-driven flow through the LC channel. Reservoir 2 was filled with the buffer solution to form the ionic contact bridge between the nanopore and LC channel. At first, a single target-carrying microbead was tapped under the nanopore and the DNAs were released from the bead using thermal heating. Following the thermal heating, an electrical potential was applied across reservoirs 1 and 2 and a very high rate of translocation was observed. As more beads were trapped, a corresponding linear increment in the capture rate was seen. Following this principle, the authors have demonstrated an almost two orders of magnitude increase in the analyte capture rate, which in turn illustrates the efficacy of their methodology. In a separate study, the lab has demonstrated a TACRE-based amplification-free detection of SARS-CoV-2 RNAs with a capture rate enhancement of over 2000× within the entire clinically relevant concentration range from 10^4^–10^9^ copies/mL [428]. Therefore, it is probably safe to say that TACRE has particular relevance in clinical diagnostics, especially at ultra-low concentrations, paving the way toward early-stage disease detection. Apart from TACRE, several other methods exist to enhance the nanopore capture rate that are developed based on different methodologies and platforms [429,430].

Control over individual single molecules is a highly desired aspect in SM experiments. Additionally, in the case of the prolonged interrogation of a single molecule, it is expected to prohibit the presence of a second molecule while one is in the process of analysis. This necessity demands a functionality that can deliver molecules on a demand basis in a highly controllable fashion, or in other words, can act as a smart gate [277,431]. Various methods are reported for the controlled gating of single molecules such as the capturing and recapturing of single molecules [432,433], wetting–dewetting-based turning on/off of a nanopore [34,434], pulsed-DC-source-based turning on/off of the pore with an adjustable sub-Hz frequency [351,379], etc. However, Rahman et al. have demonstrated a smart way of delivering single molecules based on real-time translocation detection with programmable reconfigurable settings [275]. The methodology was developed on the ARROW optofluidic platform, which is depicted in Figure 8d(i) [275]. As electronic control provides the most efficient and high-speed control over processes, the authors used a microcontroller-based real-time feedback control mechanism over the nanopore. Similar to the previous example, a nanopore was drilled on the LC channel using a dual-beam electron microscope, and a total of three reservoirs were placed on the device, two at the ends of the LC channel and one on top of the nanopore. This time, the analyte samples were loaded on the nanopore reservoir (reservoir 2), and a voltage was applied across reservoirs 1 and 2 to translocate analytes from the nanopore reservoir to the LC channel through the nanopore. As described earlier, as an analyte passes through the nanopore, it creates a spike in the current signal. In this particular case, the microcontroller continuously monitors the current signal in real-time and looks for spikes, or in other words, analyte translocation. With proper settings, it is possible to turn off the applied voltage across the nanopore (voltage gating) using a solid-state relay and relevant circuitry, therefore delivering a single analyte to the LC channel while at the same time prohibiting further analyte translocation. As a demonstration of the principle, the authors have shown a voltage-gated delivery of a single 70S ribosome where the voltage across the nanopore (red trace) was turned off just after the molecular translocation, as depicted in Figure 8d(ii) [275]. In this study, the authors have demonstrated the voltage-gated delivery of a variety of biomolecules such as DNAs, ribosomes, proteins, and NaCMC molecules, thereby proving the versatility and compatibility of their methodology with a wide range of biomolecules. In contrast to delivering just one analyte, it is also possible to deliver any desired number of analytes to the LC channel for subsequent interrogation. Additionally, the voltage across the nanopore was automatically re-applied after a certain time interval following the user-defined settings, which can be adjusted to any desired value based on the experimental requirement. The methodology is fast enough to allow for the delivery of biomolecules in rapid succession. The authors have demonstrated the successive delivery of ribosomes at rates of several hundred/min with a device limit nearly up to the kHz range, which paves the way towards a high-throughput single molecule delivery and analysis platform. Furthermore, as different biomolecules usually produce distinguishable translocation patterns based on their physical structure, charge, etc., it is possible to identify and distinguish molecules by analyzing the translocation spikes [275]. In this case, the gating platform was capable of analyzing the translocation spikes in real-time and making decisions based on the predetermined criteria. In individual control experiments, it was observed that the ribosomes produced translocations with a longer duration (dwell time) and differential amplitudes compared to the λ-DNAs when separately passed through the pore. Based on the mentioned criteria, they have demonstrated selective voltage gating when the DNA molecules translocated through the pore, while not gating when the ribosomes translocated through the pore, as shown in Figure 8d(iii) [275]. Furthermore, the feedback gating functionality can be integrated with other SM methods which have been demonstrated by combining optical methods with the feedback gating platform for simultaneous electro-optical detection of single λ-DNA. It is also possible to combine the feedback gating with other SM methods as well such as on-chip trapping, which can enable a prolonged analysis of a single particle on a demand basis. Therefore, this methodology demonstrates an elegant way of precisely monitoring, controlling, and delivering of single molecules in real-time towards developing a true single-molecule platform. Apart from electrical control and manipulation, a number of other methods are also reported to control and manipulate (mixing, size filtering, particle manipulation, etc.) single molecules [435,436].

With the advancement in micro/nanotechnology, the interest in miniaturized lab-on-a-chip devices is increasing in rapid succession. These lab-on-a-chip devices are performing unprecedented levels of laboratory bench-scale tasks within a single chip. Recently, there has been extensive research and development to add more functionalities and capabilities to these lab-on-a-chip devices such as on-chip laser, on-chip sample preparation, etc., toward developing a more complete platform for SM studies. A number of efforts are reported to develop on-chip lasers sources [437,438,439,440] that can ease the optical integration and potentially eliminate the complex optical alignment required to carry out the SM studies. Significant advances have been made toward on-chip sample preparation [271,441,442,443] for developing the next-generation point-of-care (POC) detection system [444,445,446,447,448]. Hybrid optofluidic integration that combines PDMS and silicon-based optofluidic devices is also opening avenues and allowing more flexibility, reconfigurability, and adaptability for SM studies [441]. With these integrated lab-on-a-chip capabilities, researchers have been able to perform a complete and thorough analysis of cells and single molecules [449,450,451,452,453], and yet many more are to come in the near future with added functionalities and enhanced capabilities.

## 4. Conclusions and Outlook

This review starts with the fundamentals of single-molecule studies and gradually builds up to the state-of-the-art applications and future trends. The study and analysis of single molecules are a broad context with a wide range of multidisciplinary involvement and applications. Compared to conventional ensemble measurements, single-molecule methods provide a wealth of information on the molecular level with finesse details and great precision, thereby becoming the method of choice for biological studies, life science, and other related disciplines. Optical and electrical methods are two widely used single-molecule methods that have gained particular attention and are vastly used both in fundamental studies and critical applications. Optofluidic platforms are probably one of the best choices for single-molecule studies as they fuse optical, microfluidic, electrical, and other technologies in a single device, being like a whole package in a single device. Integrated optofluidic devices are being used for the proper understanding of many complex processes at the molecular level, to address fundamental biological questions, and to gain an unprecedented level of access to control and manipulate individual molecules which have a significant impact on life science, biology, diagnostics, medical goods, healthcare, and numerous other disciplines. Due to its proven potential, several optofluidic companies are already commercializing their products, and some of them are on the verge of commercialization; once mature, they are expected to make a substantial contribution. Along with the emerging technologies, more advanced integration with these platforms is imminent toward a more complete platform with added capabilities that can perform more meaningful tasks. With extreme sensitivity, a precise spatiotemporal resolution, and high throughput, there is every reason to believe that single-molecule techniques have just sparked the paradigm shift and are on their way toward the ultimate development and reshaping of the related fields.

## Figures and Tables

**Figure 1 micromachines-13-00968-f001:**
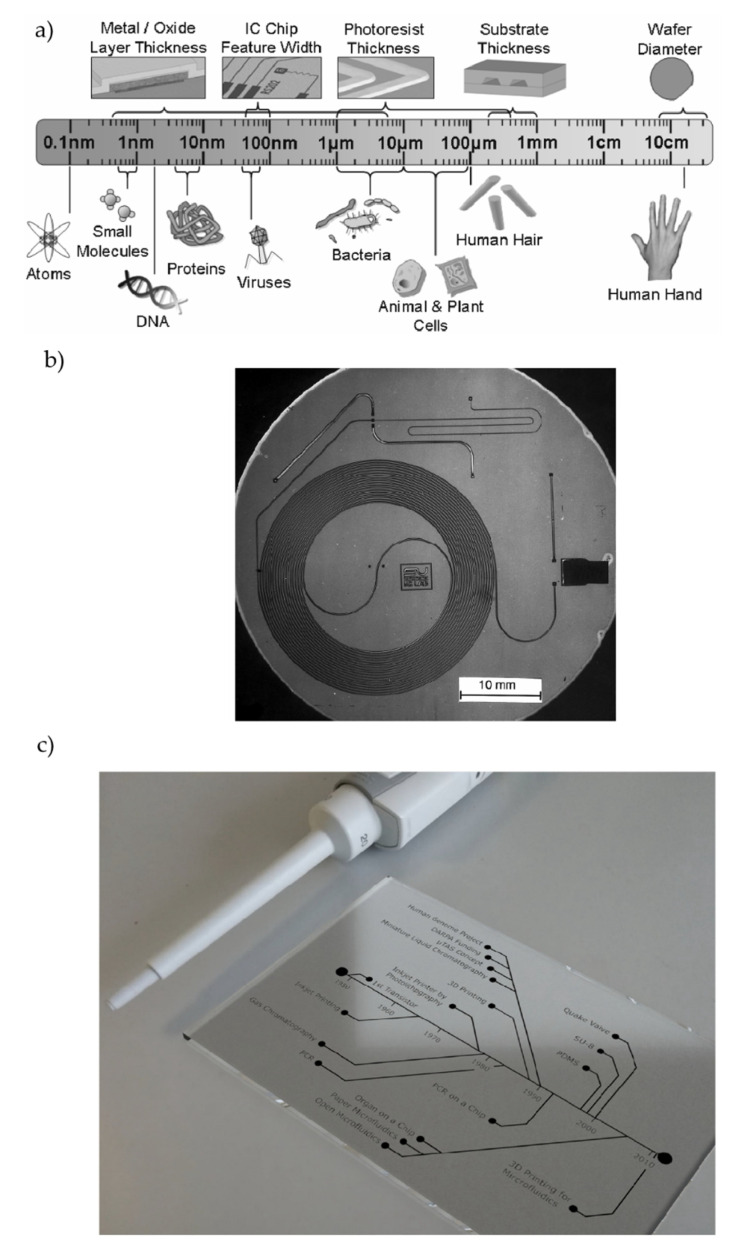
(**a**) Approximate length scales for several biological and micro fabrication structures [50]. (**b**) Photograph of a gas chromatograph integrated on a planar silicon wafer fabricated by Terry and co-workers at Stanford University [59]. (**c**) Timeline highlighting the main advances in the field of microfluidics starting with the invention of the transistor and leading up to the rise in 3D-printed devices [62].

**Figure 2 micromachines-13-00968-f002:**
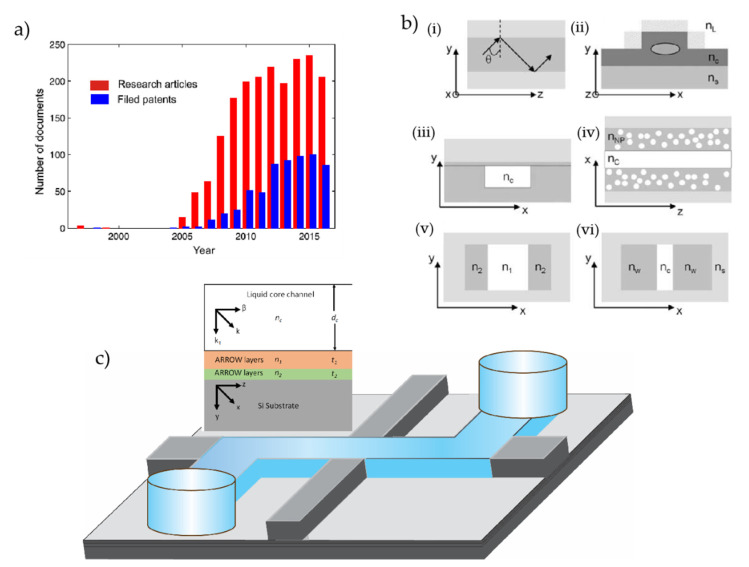
(**a**) The numbers of published research articles and filed patents in chronological order in the domain of optofuidics (data retrieved from Scopus on 26 May 2017) [119]. (**b**) Total internal reflection (TIR)-based waveguides [121]. (**i**) TIR principle in slab waveguide, (**ii**) cross section of solid core ridge waveguide with mode area (ellipse) and penetration area into surrounding liquid (hatched areas); (**iii**) liquid-core waveguide (LCW) cross section, (**iv**) nanoporous cladding waveguide (side view), (**v**) liquid–liquid core (L^2^) waveguide (cross section); (**vi**) slot waveguide (cross section). (**c**) Schematic representation of a typical ARROW optofluidic device with the wave vectors ((**c**) is used upon permission from the Applied Optics group at the University of California, Santa Cruz, CA, USA).

**Figure 4 micromachines-13-00968-f004:**
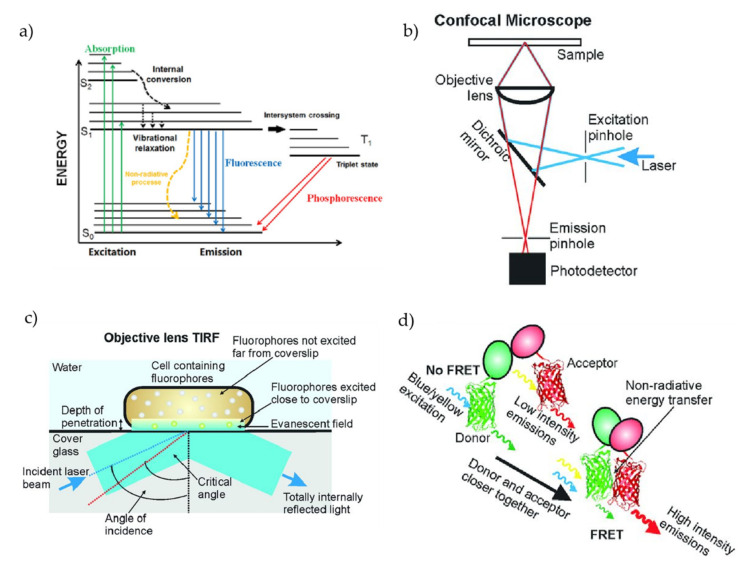
(**a**) Perrin Jablonski diagram of fuorescence and phosphorescence [232]. (**b**) Schematic representation of a confocal microscope [233]. (**c**) Schematic representation of TIRF showing the illumination of fluorophores close to the glass coverslip surface [233]. (**d**) Schematic representation of the FRET principle based on the non-radiative energy transfer which occurs when donor and acceptor dyes pair [233].

**Figure 6 micromachines-13-00968-f006:**
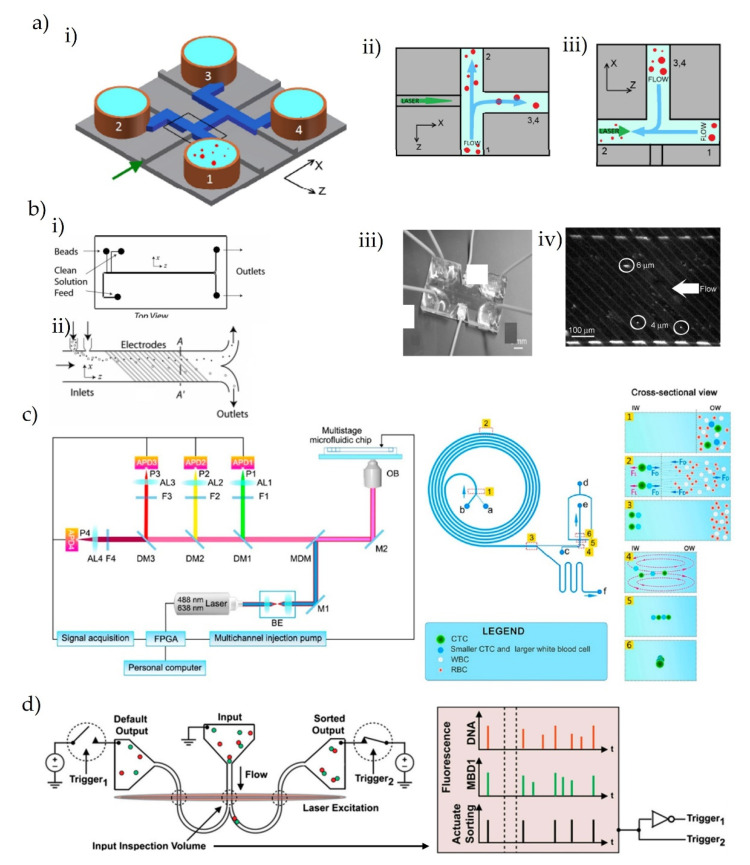
(**a**) (**i**) Layout of the ARROW optofluidic sorting device [283]. (**ii**,**iii**) Orientation of laser and flows [283]. (**b**) (**i**) Schematic of the sorting device with three inlets and two outlets [288] (**ii**) A cross-sectional view of the flow channel shows the locations of the planar electrodes and the beads during operation [288]. (**iii**) Photograph of the actual device [288]. (**iv**) Snapshot of 4- and 6-ím particles after separation at a distance of 30 mm from the inlet [288]. (**c**) Schematic Diagram of the Optical Path of the Four-Color Fluorescence Detection System and the Overall Composition of the OFCM. BE, Beam Expander Collimator; M1, M2, Reflectors; MDM, Multiband Dichroic Mirror; OB, Objective; DM1–DM3, Dichroic Mirrors; F1–F4, Filters; AL1–AL4, Lens; P1–P4, Pinholes; APD1–APD4, Avalanche Photodiodes; FPGA, Field-Programmable Gate Array [306]. (**d**) Single molecule detection and sorting [315].

**Figure 7 micromachines-13-00968-f007:**
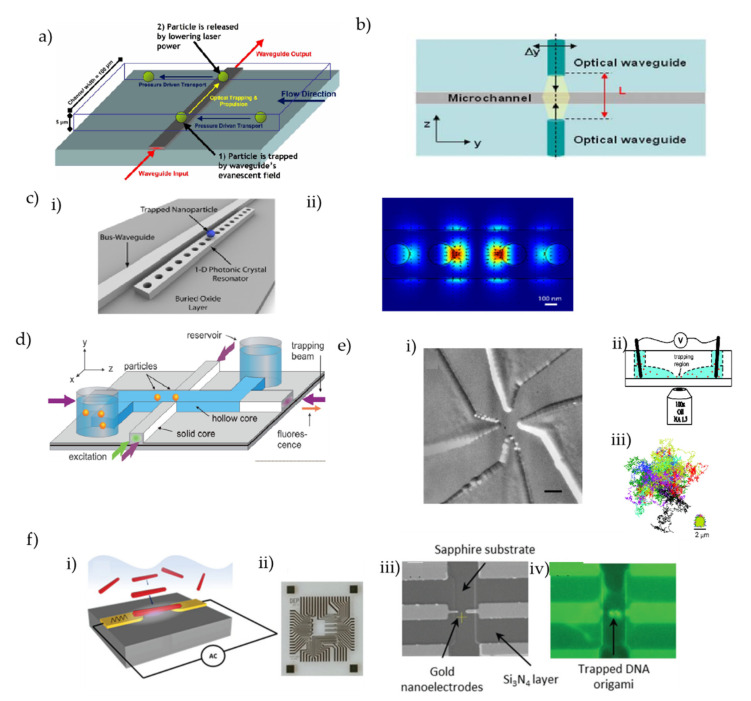
(**a**) Schematic of trapping experiment. The optical waveguide propulsion is perpendicular to the direction of the pressure-driven flow in the channel [333]. (**b**) Basic scheme of the optofluidic chip: the two waveguides emit counter-propagating Gaussian beams. The sample under testing flows into the microchannel [334]. (**c**) (**i**) 3D schematic of the one-dimensional photonic crystal resonator optical trapping architecture [337] (**ii**) 3D FEM simulation illustrating the strong field confinement and amplification within the one-dimensional resonator cavity [337]. (**d**) Schematic layout of optofluidic loss-based trapping and manipulation of particles, showing intersecting solid- and liquid-core ARROW waveguides and relevant optical beam paths [279]. (**e**) (**i**) Glass microfluidic cell for the ABEL trap with the trapping region showing the patterned glass cell [338]. (**ii**) The microfluidic cell sits above the oil-immersion objective of an inverted optical microscope capable of observing single molecules [338]. (**iii**) Trajectories of 13 trapped particles of TMV [338]. (**f**) Dielectrophoretic chip for trapping DNA origami [341]. (**i**) Device schematic structure of a dielectrophoresis chip on a sapphire substrate. Generated AC electric field in between gold nanoelectrodes absorbs DNA origamis to the high-intensity region of the electric field and immobilizes it in a specific direction by binding the thiol linkers of DNA origami to gold electrodes; (**ii**) Optical image of a fabricated chip on a sapphire substrate including 14 devices; (**iii**) SEM image of a typical device on the chip showing gold nanoelectrodes and nitride encapsulation; and (**iv**) Fluorescent image of a device with DNA origamis trapped between nanoelectrodes.

**Figure 8 micromachines-13-00968-f008:**
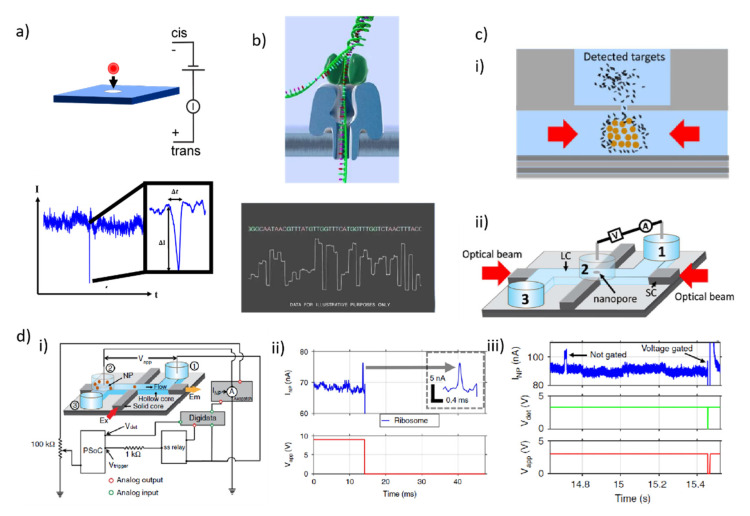
(**a**) Schematic illustration of the nanopore working principle and detection signal [14]. (**b**) The principle of nanopore-based nucleic acid sequencing [389]. (**c**) Trap-assisted capture rate enhancement of a nanopore [422]. (**i**) A cartoon depicting the conceptual visualization of TACRE. (**ii**) Schematic illustration of the experimental setup. (**d**) On-demand target delivery on a programmable ARROW optofluidic device [275]. (**i**) Schematic illustration with feedback control mechanisms. (**ii**) Current (**top**) and voltage (**bottom**) trace of a voltage gated single ribosome delivery. (**iii**) Current (**top**), identification signal (**middle**), and voltage (**bottom**) trace of identification and voltage gating of only λ-DNAs from a mixture of λ-DNA and ribosome.

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
