# Peer review of "A Critical Review on the Sensing, Control, and Manipulation of Single Molecules on Optofluidic Devices"

_micromachines, 2022, doi:10.3390/mi13060968_

Round 1

Reviewer 2 Report

  1. The content of each section suggests adding some new pictures and further segmentation to increase the readability of the article.
  2. There is not much description of the progress of various applications. It is suggested that the author describe the progress in combination with some pictures, and pick out one or two representative works or the latest technical scheme.
  3. The advantages and disadvantages of each application should not be simply described. It is suggested that the author can further describe its possible solutions for their disadvantages.
  4. There is a problem with the format of references 34 and 35.

Round 2

Reviewer 1 Report

I am satisfied with this revision and it can be ready to publish with some given insights on the challenges of SM technique in the conclusion section.